# Olympic Italian Female Water Polo Players: Analysis of Body Size and Body Composition Data over 20 Years

**DOI:** 10.3390/jfmk10020210

**Published:** 2025-06-04

**Authors:** Giovanni Melchiorri, Marco Bonifazi, Maria Rosaria Squeo, Raffaella Spada, Virginia Tancredi, Valerio Viero

**Affiliations:** 1School of Sport and Exercise Sciences, Department of Systems Medicine, Faculty of Medicine and Surgery, “Tor Vergata” University of Rome, 00133 Rome, Italy; gmelchiorri@libero.it; 2Don Gnocchi Foundation IRCCS, 20162 Milan, Italy; 3Italian Swimming Federation, 00135 Rome, Italy; marco.bonifazi@unisi.it; 4Department of Medical Biotechnologies, University of Siena, 53100 Siena, Italy; 5Medicine and Sport Science Institute, Coni, 00135 Rome, Italy; mariarosaria.squeo@coni.it (M.R.S.); ext_raffaella.spada@coni.it (R.S.); 6Department of Systems Medicine, “Tor Vergata” University of Rome, 00133 Rome, Italy; tancredi@uniroma2.it; 7Centre of Space Bio-Medicine, “Tor Vergata” University of Rome, 00133 Rome, Italy

**Keywords:** secular trend, body mass, body height, age, nutrition, water polo

## Abstract

**Background:** The variation in the body mass and height of players over time is called the secular trend. It has been analyzed in several team sports, but no similar studies have been conducted on female athletes playing water polo. The aim of this paper was to study the changes that have occurred in the body size and composition of female water polo athletes participating in the Olympic Games, from their first inclusion in the Olympics (2004) until today. **Methods:** Data were collected from the female water polo players of the National Team selected to participate in the Olympic Games from 2004 (Athens) until 2024 (Paris) and then analyzed. A total of 93 athletes were assessed, and we analyzed the data for each of the Olympics between 2004 and 2024. To evaluate the anthropometric characteristics of the athletes, their body mass and height were recorded and their Body Mass Index (BMI) was then calculated. The athletes’ Body Composition (BC) was assessed using bioelectrical impedance analysis (BIA). **Results:** The athletes’ average age decreases over time, while their body mass increases. Their body height does not vary significantly. The BMI confirmed that the athletes were always healthy and with a correct diet. With regard to BC, the Fat Free Mass (FFM) values exhibit an increasing trend. **Conclusions:** The water polo female athletes participating in the 2024 Olympic Games were younger and have different anthropometric and BC values than the athletes playing in the first women’s water polo tournament at the Olympics in 2004. The most likely explanation for this is the rapid evolution of the young female version of the sport, with improved recruitment and training strategies and greater attention paid to nutrition.

## 1. Introduction

To analyze the evolution of performance in sports, several authors have studied the changes in the body size and composition of athletes over time [1,2]. The variation in the body mass and height of athletes over time, called the secular trend, has been analyzed in basketball, baseball and football, for which databases have been present since the early 1900s [2]. In all these team sports, a secular increase in body mass, body height and Body Mass Index (BMI) was registered [2]. In recent decades, studies on the secular trend have been conducted on different generations of young soccer players (body height and weight) [3] and young people who do or do not engage in sports (fitness levels) [4,5]. A study analyzed anthropometric data over a 15-year period but on a male water polo athletes’ sample [6]. However, no similar studies have been conducted on water polo female athletes.

Water polo is a highly demanding and complex team sport that requires repeated sprints at maximum speed interspersed with swimming at a slower speed during the game [7,8]. In addition, the technical movements associated with the sport and the need for frequent contact with the opponent, with actual fighting sometimes occurring in the water, force players to frequently switch from a vertical to horizontal position during the game [9]. Water polo therefore requires many hours of training; up to 35 h a week [10]. Athletes must be prepared for all technical and tactical components of the game and be able to perform the required skills appropriately in optimal physical condition. In addition, to achieve the required level of performance, gym training with heavy loads is undertaken, while training in the water is strenuous and prolonged [10].

In 2004, the Italian Olympic team won the Olympics. Since then, it has achieved numerous important international results, being competitive in all the main events, which proves how widespread water polo has become in Italy and reveals the good technical level of Italian national championships. Considering the evolution in performances, we wanted to investigate the changes undergone by our athletes over time.

In the analysis of the physical characteristics of athletes, their body composition [2,6,10] has been studied for many years now, as it enables researchers to obtain more in-depth information about athletes’ physical characteristics (fat free mass, fat mass, hydration, phase angle) compared to simple body mass. Weight loss and dehydration can represent symptoms and signs of Non-Functional Overtraining and Over-Training Syndrome [10]. In water polo, the use of bioelectrical impedance analysis (BIA) is widespread [10], but no longitudinal studies have evaluated the change in the body composition of athletes over the years, at least until we last checked the reference material on the PubMed database in September 2024.

Studying the essential body size (body mass and height) of athletes and those related to body composition in sports is certainly of interest considering that the evolution of performance is also due to the improvement of athletes’ physical characteristics. The modification of these characteristics is attributable to natural human evolution, to selection based on variations in the features of games, to nutritional aspects, to the improvement of training methodologies, and to psycho-social aspects that have made the athletes more aware and attentive to the management of their talent [2]. Further studies have confirmed the importance of Body Composition (BC) in different sports and sport positions when determining the differences between athletes [11], while others have investigated aspects related to BC in women’s water polo via techniques other than BIA [12].

Over the last 20 years, in sport in general and especially in team sports, there has been a strong increase in the number of female participants [13]. Due to the specific physiology associated with the female body, the changes that these women likely undergo over time (for example, pregnancy), and the more recent increase in the number of female water polo athletes compared to men (first appearance of female water polo at the Olympic Games in 2004), it is interesting to study variations in the physical characteristics of female athletes [10].

Therefore, the aim of this study was to analyze the changes that have occurred in the body size and composition of Italian female water polo athletes participating in the Olympic Games, from their first inclusion in the Olympics (2004) until today. The research hypothesis was that similarly to other sports, body mass and body height increased over the examined period.

## 2. Materials and Methods

### 2.1. Subjects

For the purposes of this study, data were collected from the female water polo players on the Italian National Team selected to participate in the Olympic Games from 2004 (Athens) until 2024 (Paris), and then analyzed. A total of 93 athletes (*n* = 93) were assessed after being divided into six groups, one for each of the Olympic Games between 2004 and 2024. The assessments were performed in the six months preceding the event. Athletes participating in more than one Olympic Game were included in the study only in their first participation. The average age of all the athletes considered was 26.2 ± 4.3 years. In the year preceding the Olympic Games, all the athletes had trained regularly with their respective clubs, played matches in their championships, and attended periodic meetings with the National Team. In the case that a significant injury resulting in the loss of more than 14 days of training occurred, the subject would be excluded from the study.

### 2.2. Experimental Procedure

Athletes who are candidates for the Olympic Games are subjected to specific medical and non-medical tests. In Italy, the Olympic Committee includes an analysis of their BC via BIA; this allowed us to retrieve all the data collected just approximately six months before the Olympics. All the athletes subjected to pre-Olympic medical examinations gave consent to anonymous data processing for scientific research purposes. Almost all of the authors are members of the Italian Swimming Federation and the Italian National Olympic Committee, both of which approved and authorized the data access and analysis. To ensure the absence of bias caused by the use of different software, the resistance and reactance data were reprocessed using Campa formulas that were specific to this sample [14].

This study was approved by the Internal Review Board of the University of Tor Vergata, with number 135 and dated 6 October 2024, and it followed the guidelines of the 1964 Declaration of Helsinki and subsequent amendments. The assessments carried out on the athletes focused on anthropometric and body composition measurements.

#### 2.2.1. Anthropometric Measurements (Body Mass, Body Height, and Body Mass Index)

To evaluate the anthropometric characteristics of the athletes, their body mass and height were recorded; then, their BMI was calculated. Body mass was measured to the nearest 0.1 kg on an electronic beam scale (Invernizzi, Rome, Italy), without clothes and shoes and after the participants had defecated. In all athletes involved in the study, two consecutive measurements of body weight were performed and in case of a difference between the first and second measurement, a third measurement was performed. In case of a difference, the recorded weight was the average value of the three measurements.

In our sample, the difference between the first and second measurement was very low (effect size: 0.02), not statistically significant (*p*: 0.63) and the repeatability between the tests was high (r: 0.99). A third measurement was used in 22% of the sample. The average difference between the first measurement (71.5 kg ± 10.0 kg) and the second measurement (71.7 kg ± 10.2) is very small.

Body height was measured to the nearest 0.5 cm using a Harpenden stadiometer (Holtain, Pembrokeshire, UK). All data were collected by the same operator using the same measurement protocol and equipment. BMI was calculated as body mass (kg)/body height (m)^2^.

#### 2.2.2. Body Composition Analysis

The BIA measurements were performed to assess athletes’ body composition. The use of BIA is widespread, especially in sports, due to its portability and ease of application. Its use with standardized protocols [15] provides data that are comparable with the best clinical methods. BIA also provides additional raw bioelectrical parameters that can be used to qualitatively track body composition: bioelectrical resistance and reactance are associated with the content of body fluid and cell density, respectively [14]. BIA measurements were performed in the morning after an overnight fast of at least 12 h and after 48 h of abstinence from alcohol consumption. The participants emptied their bladders within 30 min of undergoing the measurements. All measurements were performed on the dominant side, while participants lay supine on an examination table with their limbs positioned away from the trunk. Four gel electrodes were attached to defined anatomical positions on the hand, wrist, ankle, and foot [10]. The BIA measurements were performed using an Akern BIA (Florence, Italy), which applies an alternating current of 800 μA at a single frequency of 50 kHz, and Littmann adhesive electrodes (3M, St. Paul, MN, USA). BIA can measure resistance, reactance and the phase angle (PhA). All measurements were performed using the same standard procedure described above and by the same expert operator. All of the data collected from the various sessions were reprocessed using Campa’s specific formulas [14].

#### 2.2.3. Relative Age

The relative age study was carried out to verify whether recruitment parameters in relation to age changed over time. The date of birth of each player was recorded, and according to that each of them was assigned to a year’s quarter.

### 2.3. Statistics

Data were initially entered into an Excel database (Microsoft, Redmond, WA, USA), and the subsequent analysis was performed using the Statistical Package for the Social Sciences Windows, version 19.0 (SPSS, Chicago, IL, USA), with descriptive statistics consisting of the mean ± standard deviation (SD). We evaluated the normality of data using normality plots, the Kolmogorov–Smirnov test and the Shapiro–Wilk test. The homogeneity of variance was assessed using Levene’s test. The groups were compared using multifactorial univariate ANOVA considering the variables divided by the combined factor by year with the Bonferroni multiple comparisons option. Considering the sample size, standard deviation was used to calculate the pooled effect size. Cohen’s d effect size (ES) was used to study the effect size according to the formula (M1 − M2)/SDi, with M1 being the mean value of the first measurement, M2 being the mean value of the second, and SDi being the standard deviation for independent groups [16]. An effect size of 0.2 was considered small, 0.5 was considered medium, and 0.8 was considered large. The value of significance was set at 0.05. For non-normal parameters, comparisons were made with non-parametric tests. Comparisons between enumeration data were instead evaluated with the Chi^2^ test or, if there were frequencies less than 5, with the Fisher’s exact test. Furthermore, the regression analysis was conducted by determining the best fitting quadratic curves for which the equations and determination coefficients (R^2^) were reported. A value of *p* < 0.05 was considered statistically significant.

## 3. Results

Table 1 shows the values of the physical and body composition parameters recorded among the six groups selected to participate in the Olympic Games from 2004 until 2024. The groups’ average age decreased over time; body mass, on the contrary, increased. However, neither age nor body mass seemed to follow a linear trend. Body height, nonetheless, did not vary significantly, but it should be noted that the standard deviation for the last three Olympic Games was remarkably lower than that for the previous ones. BMI always fell within normal values and exhibited a low standard deviation, thus confirming, along with the other BC values, that the athletes were always healthy and of normal weight.

With regard to body composition, the FFM values exhibited an increasing trend also if relative FFM (%) did not change.

To better understand the trend over time of the variables investigated, it should be noted that between 2004 and 2024 statistically significant differences were found: Age (*p* = 0.03, ES = 0.84, ∆% = −12.0%); Body Mass (*p* = 0.05, ES = −0.77, ∆% = 8.9); FFM (kg) (*p* = 0.05, ES = −0.79, ∆% = 8.9).

Table 2 displays the values relating to the relative age of all the groups.

In Table 2 it should be noted that in both groups (2004 and 2024) over half of the athletes were born in the first half of the year. In superscript are differences showing statistical significance (*p* < 0.005): I, II, III and IV are the year’s quarter; 04 is 2004, 08 is 2008, 12 is 2012, and 21 is 2021 Olympic Edition.

Figure 1 shows the trend in body mass over time from 2004 to 2024.

Figure 2 shows the increase in FFM over time.

Table 3 shows some statistically significant differences appreciable in absolute values only (increase in ICW, ECW, and TBW) between 2004 and 2024. Considering the differences in body mass recorded over time, we also reported the data as relative values. Percent value presents no statistically significant difference, which proves that the variations in ICW, ECW, and TBW volumes, are underpinned by the increase in body mass. Data of hydration in the other groups (2008, 2012, 2016 and 2021) were substantially the same. Different hydration, as in the case of dehydration, could also have affected the BIA measurements. The absence of statistically significant variations allows for a more reliable comparison between data obtained 20 years apart.

## 4. Discussion

The aim of this study was to analyze the changes in the body size and composition of female water polo athletes from their first inclusion in the Olympic Games (2004) until today. The research hypothesis was confirmed: the athletes of the 2024 group were younger and had a significant increase in body mass and in FFM. They were also taller than the participants in the Olympics in 2004 but the value was not statistically significant.

A first relevant fact concerns the age of the sample analyzed. The study showed that the age of the athletes significantly decreased (*p* = 0.03) between the 2004 and 2024 Olympic Games; this suggests that athletes are now reaching their sport’s top level and therefore the most important competition, namely the Olympic Games, at a younger age. A possible explanation for this phenomenon is the recruitment of young athletes, leading to the optimization of their sports career. In fact, Viero and associates have suggested that it is important to detect talent at its optimum so as not to waste or delay the manifestation of individual talent [17]; thus, even very young children are introduced to the sport. This criterion sometimes goes too far and leads to the implementation of a too early specialization, which does not take into account the actual needs of the young athlete and can result in negative effects [18]. The improvement of training systems and the greater dissemination of knowledge regarding nutrition may also have led to this change [19]. The younger ages could be also interpreted as the beginning of a new cycle (renovation).

As we can see in Table 2, there is an appreciable change trend of relative age value over the years (first year’s quarter 2004 = 14.3% and 2024 = 26.7%), and over half of the athletes in both groups (2004 and 2024) were born in the first half of the year (2004 = 57.3% and 2024 = 60%) but no statistical significance was found.

These values show that younger athletes tend to be selected and that even if the data are not statistically significant attention must be paid to a possible effect of relative age. Alongside the decrease in age observed in the samples analyzed, we recorded a significant increase in body mass (*p* = 0.05). While such a result could be expected due to the effects of the secular trend on the general population, it is important to note that this increase is very significant and exceeds expectations; this is especially considering that the effects of the secular trend are gradually diminishing, particularly in more developed countries [1].

In American male athletes participating in various team sports, an average increase in body mass has been recorded over several decades; this ranges from 1.5 kg per decade for ice hockey players to 2.7 kg per decade for football players [2]. In our sample of female athletes, this increase was over 3 kg per decade. In the only article on the secular trend in water polo Lozovina did not detect any significant changes in body mass but his study dealt with male athletes in a period before our research (1980–1995) [6]. The secular trend in body mass and height seems to have greater effects on athletes than on the normal population, due to the different criteria that have developed over time due to changes in the rules and features of sports [1], particularly recruitment and training criteria. For example, over the years, the body height of elite basketball players has increased significantly more than that of athletes participating in other sports; meanwhile, the body mass of football players has increased much more [1]. This explanation is even more likely when considering relatively young, and therefore constantly evolving, sports such as women’s water polo. Muscle strength training, for example, which can lead to an increase in body mass, is now much more widespread in elite women’s water polo than it was twenty years ago. Furthermore, research studies on the talent detection criteria used in water polo highlight the importance of body mass and muscular strength [17]. Improved knowledge and a broadened culture of correct nutrition have also been shown to play an important role in the body composition and physical performance levels of athletes participating in team sports [19].

Furthermore, there have been no significant changes in the body height of these athletes between the Athens (95% CI; 167.1–175.9) and Paris (95% CI; 171.0–174.9) Olympic Games, although an increase of 1.4 cm in their mean body height (0.7 cm/decade) was recorded in the twenty years analyzed. In American male hockey, baseball and football players, the average increase in body height was found to be 0.9 cm/decade over much longer periods of time (9–14 decades) [2]. In our data, the standard deviation shows a different distribution, with there being a higher number of taller athletes in the Paris Olympic Games compared to the Athens Olympic Games. This could also be due to different recruitment strategies based on technical needs.

The BMI values confirm an optimal body mass in relation to body height and show an increase consistent with that of body mass. We only considered BMI in this context because, in athletes, it must be used with caution; this is due to the greater muscle mass of athletes and the use of criteria that are different to those used in the normal population [20].

We compared the data collected among elite female water polo players in 2024 with those available for the normal population, dating back to 2020 [21]. The average body mass was 64.8 kg in the normal population, but 74.3 kg among athletes (∆ = +14.7%). These values, compared with those obtained by Cacciari and associates in 2006 for women of the same age, are in the 80° and 96° percentile, respectively [22]. In addition, the body height of the athletes was 164.6 cm, compared to 173.0 cm (∆ = +5.1%; 55° and 97° percentiles, respectively), while their BMI was 24.9, compared to 24.8 (∆ = −0.40%; both in the 84° percentile).

With regard to the FFM values (Table 1), we observed a significant increase (*p* = 0.05) due to the body mass increasing since relative FFM (%) does not change; this probably indicates that there was an increase in the need for muscle strength in competition and that muscle conditioning between the Athens (95% CI; 49.9–55.5) and Paris (95% CI; 53.4–63.1) Olympic Games has maintained great importance. Undoubtedly, the characteristics of the game, which is extremely challenging in itself and increasingly demanding in terms of power, speed and rhythm, have led coaches to select athletes according to different criteria (more body mass) and to enhance their performance through training strategies that aim to enhance their physique (strength training) [17] and provide adequate nutritional support.

The PhA does not significantly change while the body composition data (FFM) show variance. The PhA is to be considered as a raw bioelectrical variable of the BIA (relation between impedance and reactance) while FM, and FFM with the BIA are calculated using predictive mathematical formulas within which other anthropometric variables are inserted in addition to the bioelectrical ones (BIA derived parameters) which could explain this difference. PhA has been studied in sports [23] to investigate fatigue, inflammatory state, training effect, type of exercise; however, we were unable to find any long-term longitudinal studies. The reported data show no significant changes in PhA values over a period of 20 years for highly competitive and well-trained water polo players. Since it has been hypothesized that the PhA value could be influenced by different factors such as ECW, ICW, ECW/ICW ratio, TBW, body cell mass, cellular integrity, age, minerals content, performance level and sport practiced [24], it can be thought that since no significant variations in ECW and ICW were recorded, the PhA value also did not change significantly. Considering the variability of the measurement [24] and the multifactorial nature that could contribute to determining the PhA value [24,25,26], new research is needed to confirm this hypothesis. According to some authors [24], the PhA could provide important information regarding the relationship between extracellular and intracellular water, body cell mass and cellular integrity. The PhA changes according to age and gender, and is higher in athletes [25]. However, among athletes participating in different sports, this bioelectric impedance value is known to be highly variable. To this day, there are still doubts about the relationship between PhA and sports performance, as well as the influence of training or non-training periods on PhA [26]. As has been suggested, to better interpret the results relating to PhA, it is necessary to understand how PhA varies in different sports and verify the changes and trends over time [26]. To date, ours is the first and only study to define the trend in PhA in a specific population of highly competitive water polo athletes. To better understand the variability in PhA, more detailed research is needed in order to analyze the values of PhA under training and non-training conditions (short and long) among this specific population.

The hydration values also confirm the good level of nutrition in the two groups (Table 3) [10]. Indeed, although the overall data regarding the Extra Cellular, Intra Cellular and Total Body Water (lt) seem significantly different at first, the differences concerning the analysis of body mass hydration (%) are insignificant; this indicates that the hydration of the two groups is similar. In the other groups of the Olympic Games hydration data were very similar, thus confirming these considerations. Different hydration conditions could also have affected the BIA measurements. The absence of statistically significant variations allows for a more reliable comparison between data obtained.

The limitations of this study are that the athletes examined were of a single nationality and that it is impossible to perform a comparison with similar articles on women’s water polo because our study seems to be the first on this topic. A comparison with articles dealing with the secular trend and not including data on body composition is also impossible. In fact, other authors have estimated the changes in the body mass and height of athletes over long periods of time, but only in men [6]. In any case, our data can be used as a reference for athletes in the Italian population. The results of this study may provide indications to female water polo technicians in talent detection and selection.

## 5. Conclusions

In conclusion, the female water polo athletes participating in the 2024 Olympic Games are younger and have anthropometric and body composition values that are more suitable for the current game than the athletes playing in the first Italian women’s water polo tournament at the Olympics in 2004. The most likely explanation for this is the rapid evolution of this young female version of the sport, with improved recruitment and training strategies and greater attention paid to nutrition. The existence of various international competitions for young female water polo players (European, World, etc.) could be seen as further evidence of this evolution.

In the future, it would be useful to extend this research internationally, create benchmark values that can be used by coaches in the selection process, and reproduce the same study on male water polo players.

## Figures and Tables

**Figure 1 jfmk-10-00210-f001:**
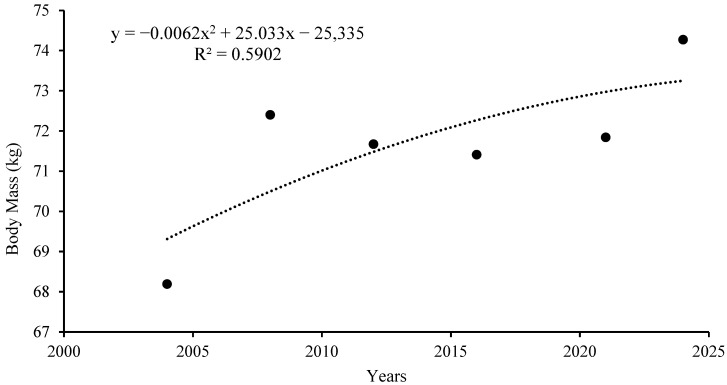
Trend in body mass.

**Figure 2 jfmk-10-00210-f002:**
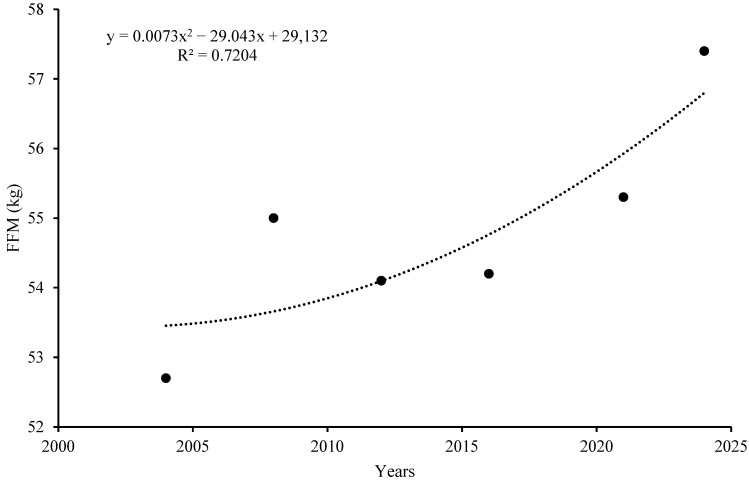
Trend in FFM (Fat Free Mass).

**Table 1 jfmk-10-00210-t001:** Sample characteristics.

Olympic Edition	Athens 2004	Beijing 2008	London 2012	Rio 2016	Tokyo 2021	Paris 2024
Sample number (*n*)	14	15	17	16	16	15
Age (yrs)	27.6 ± 4.0 ^(P)^	27.8 ± 3.6 ^(P)^	24.9 ± 5.3 ^(P)^	25.1 ± 4.1	27.4 ± 3.3 ^(P)^	24.3 ± 4.1 ^(A-B-L-T)^
Body height (cm)	171.6 ± 7.6	174.3 ± 7.1	173.0 ± 7.9	172.3 ± 3.9	172.7 ± 3.7	173.0 ± 3.5
Body Mass (kg)	68.2 ± 7.0 ^(P)^	72.4 ± 12.4	71.7 ± 14.8 ^(P)^	71.4 ± 5.5	71.8 ± 9.1	74.3 ± 9.2 ^(A-L)^
BMI	23.2 ± 2.6	23.8 ± 3.2	23.8 ± 3.4	24.0 ± 1.5	24.1 ± 3.0	24.8 ± 3.0
FM (kg)	15.5 ± 2.3	17.4 ± 4.5	17.5 ± 4.6	17.2 ± 1.9	16.5 ± 3.7	16.8 ± 3.0
FM (%)	22.6 ± 1.4	23.7 ± 2.2	24.3 ± 2.1	24.1 ± 2.1	22.8 ± 2.9	22.7 ± 2.4
FFM (kg)	52.7 ± 4.9 ^(P)^	55.0 ± 8.1	54.1 ± 10.6 ^(P)^	54.2 ± 4.6 ^(P)^	55.3 ± 6.1 ^(A-P)^	57.4 ± 7.1 ^(A-L-R-T)^
FFM (%)	77.4 ± 1.4	76.3 ± 2.2	75.7 ± 2.1	75.9 ± 2.1	77.2 ± 2.9	77.3 ± 2.4
PhA	7.5 ± 1.2 ^(L)^	7.3 ± 0.5 ^(L)^	6.8 ± 0.5 ^(A-L-T-P)^	7.6 ± 0.8 ^(L)^	7.9 ± 0.6 ^(L)^	7.7 ± 0.7 ^(L)^

Results are shown as mean ± standard deviation (SD). BMI, Body Mass Index; FM, Fat Mass; FFM, Fat Free Mass; PhA, Phase Angle. The table shows the data recorded over the analyzed span of years. Statistical significance for each variable was estimated by comparing the value of the same variable recorded in the various years. Statistical significance was set with a *p* value < 0.05. Where a statistically significant difference was found, a letter next to the value means the existence of a statistically significant difference from the value of the same variable measured on another date (P: Paris 2024; T: Tokyo 2021; R: Rio 2016; L: London 2012; B: Beijing 2008; A: Athens 2014).

**Table 2 jfmk-10-00210-t002:** Relative age.

Year’s Quarter	I	II	III	IV
2004 n14	14.3%	42.9% ^I 21^	21.4%	21.4%
2008 n15	13.3%	40.0% ^I 21^	13.3%	33.3%
2012 n17	5.9% ^III 21; III,IV 12^	11.8%	41.2% ^I 12; I 21^	41.2% ^I 12; I 21^
2016 n16	18.8%	18.8%	25.0%	37.5%
2021 n16	6.3% ^II 04; II 08; III,IV 12^	37.5%	31.3%	25.0%
2024 n15	26.7%	33.3%	26.7%	13.3%

**Table 3 jfmk-10-00210-t003:** Hydration considering intra and extra cellular water.

	ECW (lt)	ECW (%)	ICW (lt)	ICW (%)	TBW (lt)	TBW (%)
2004	16.2 ± 1.1	23.8 ± 1.0	21.8 ± 2.3	32.0 ± 0.6	38.0 ± 3.4	55.7 ± 1.0
2024	17.3 * ± 1.7	23.3 ± 0.8	23.9 * ± 3.2	32.2 ± 0.9	41.2 * ± 4.9	55.5 ± 1.5
p	0.05	0.20	0.05	0.41	0.05	0.67
ES	0.8	0.6	0.8	0.3	0.8	0.2
∆	1.1	−0.5	2.1	0.2	3.2	−0.2
∆%	6.8	−2.1	9.6	0.6	8.4	−0.4

The data reported in the table refer to Total Body Water (TBW), Extra Cellular Water (ECW) and Intra Cellular Water (ICW). The variables are expressed in absolute terms (lt) and in relative terms (percent value, %, in relation to body mass). * statistically significant difference. Results are shown as mean ± standard deviation (SD).

## Data Availability

All data from the study are included in the present manuscript.

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
