# Peer review of "Olympic Italian Female Water Polo Players: Analysis of Body Size and Body Composition Data over 20 Years"

_jfmk, 2025, doi:10.3390/jfmk10020210_

Round 1
Reviewer 1 Report (New Reviewer)
Comments and Suggestions for Authors
I SUGGEST MAJOR REVISIONS (23 comments). See pdf. file

Author Response
Reply to reviewer #1
- Authors’ addresses: ACCEPTED: authors’ email inserted.
Abstract:
- Delete: ACCEPTED: sentence deleted. “Abstract: Background/Objectives: Over the last 20 years, in sport in general and especially in team sports, there has been a significant increase in the number of female practitioners.”
- Rephrase the sentence: ACCEPTED: sentence modified. “The variation in the body mass and height of players over time is called the secular trend. It has been analyzed in several team sports, but…”
- Paper: ACCEPTED: work has been changed in paper. “The aim of this paper was to study the changes…”
Keywords:
- Add keyword water polo: ACCEPTED: keyword water polo added.
Introduction:
- Tell us more about…: ACCEPTED: a sentence has been added. “In all these team sports a secular increase in body mass, height and Body Mass Index (BMI) was registered [2].”
- This text should be line…: ACCEPTED: the text has been moved.
- This sentence provides…: ACCEPTED: the sentence has been divided in two sentences. “Studying the essential body size (body mass and height) of athletes and those related to body composition in sports is certainly of interest considering that the evolution of performance is also due to the improvement of athletes’ physical characteristics. The modification of these characteristics is attributable to natural human evolution, to selection…”
- Therefore, the aim…: ACCEPTED: sentence modified. “Therefore, the aim of this study was to analyze the changes…”
- Establish a research hypothesis…: ACCEPTED: a research hypothesis has been added. “The research hypothesis was to find an increase in body mass and height like in other team sports [2].”
Methods:
- (n=93): ACCEPTED: sentence modified. “A total of 93 athletes (n=93) were assessed…”
- Reformulate…: ACCEPTED: sentence modified. “Athletes who are candidates for the Olympic Games are subjected to specific medical and non-medical tests. In Italy, the Olympic Committee includes an analysis of their BC via BIA; this allowed…”
Results:
- Font: ACCEPTED: corrected.
- Bold: ACCEPTED:
- Size: ACCEPTED: corrected.
Discussion:
- Clearly emphasize the main results…: ACCEPTED: two sentences have been added. “. The research hypothesis was confirmed: the athletes of the 2024 group were younger and had significant increase in body mass and in FFM. They also were taller than the participants in the Olympics in 2004 but the value was not statistically significant.”
- Viero and associates: ACCEPTED: sentence modified.
- Is early specialization…: ACCEPTED: the sentence has been modified to make the meaning clearer. “This criterion sometimes goes too far and leads to the implementation of a too early specialization, whichdoes not take into account the actual needs of the young athlete and can result in negative effects [18].”
- Without space: ACCEPTED: corrected.
- What does this mean?: ACCEPTED: the sentence has been modified to make the meaning clearer. “In our sample of female athletes, this increase was over 3 kg per decade. In the only article on the secular trend in water polo Lozovina did not detect any significant changes in body mass but his study dealt with male athletes in a period before our research(1980-1995) [6]. The secular trend in body…”
- Normal population: the text has been edited by the linguistic service of MDPI and also by an English language native. This linguistic expression is present also in other parts of the discussion.
- Cacciari and associates: ACCEPTED: sentence modified.
- Doi: ACCEPTED: references modified.
Reviewer 2 Report (New Reviewer)
Comments and Suggestions for Authors
Dear Authors
There are minor problems with the formation of the tables, and I made some suggestions.
The legend of Table 1 should be mentioned as a legend.
Se the formations of Table 2 (why only 2004 and 2024)
The title of Table 3. could be more explicit with the variables "Hydration considering intra and extra Cellular water. Also the legend.
The study could add more contributions if the nutritional assessment were available.
Note: the younger ages could be interpreted as the begining of a new cycle (renovation)
Congratulations!
Additional comments:
The authors in their study aim to analyze the changes that have occurred in the physical (Body size and composition) characteristics of Italian female water polo athletes participating in the Olympic Games, from their first inclusion in the Olympics (2004) until today. Although day use physical characteristics the more accurate should be body size and compostion, could be consider for same a matter of semantic.
The topic is relevant considering the longitudinal data provided by the authors in a specific sport where the gender of the study is female, which is scarce on published data.
The useded methodology is according to most of the studies published in this topic. However, it is missing some anthropometric variables normally used, and regarding BIA, it is unclear if the same equipment was used for all the Olympic teams and in the same conditions.
The conclusions are consistent with the presented results and answer the main question.
The used references are adequate to the topic and adequate.
The presented tables are adequate; table 2 only establishes data from 2004 and 2024, and the other years? Was it the author's choice?
Author Response
Reply to reviewer #2
Comments and Suggestions for Authors
Dear Authors
There are minor problems with the formation of the tables, and I made some suggestions.
- The legend of Table 1 should be mentioned as a legend.
ACCEPTED: modified as suggested.
- See the formations of Table 2 (why only 2004 and 2024)
ACCEPTED: the table has been completed with all the other editions of the Olympics. New sentences have been added. “As we can see in Table 2, is appreciable a change trend of relative age value over the years (first year’s quarter 2004 = 14.3% and 2024 = 26.7%), and over half of the athletes in both groups (2004 and 2024) were born in the first half of the year (2004 = 57.3% and 2024 = 60%) but no statistical significance was found.”
Table 2. Relative age.
Year's quarter I II III IV
2004 n14 14,3% 42,9% I-2021 21,4% 21,4%
2008 n15 13,3% 40,0% I-2021 13,3% 33,3%
2012 n17 5,9% III 2021, IV-2022 11,8% 41,2% I-2021 41,2% I-2021
2016 n16 18,8% 18,8% 25,0% 37,5%
2021 n16 6,3% 37,5% 31,3% 25,0%
2024 n15 26,7% 33,3% 26,7% 13,3%
In Table 2 must be noted that in both groups (2004 and 2024) over half of the athletes were born in the first half of the year. In superscript differences showing statistical significance (p< 0.005) .
- The title of Table 3. could be more explicit with the variables "Hydration considering intra and extra Cellular water. Also the legend.
ACCEPTED: modified as suggested. “Table 3. Hydration considering intra and extra cellular water.”
- The study could add more contributions if the nutritional assessment were available.
ANSWER: we totally agree but unfortunately the athletes trained in their clubs and it was not possible to have nutritional assessment.
- Note: the younger ages could be interpreted as the beginning of a new cycle (renovation).
ACCEPTED: Inserted in discussion. “…The improvement of training systems and the greater dissemination of knowledge regarding nutrition may also have led to this change [19]. The younger ages could be also interpreted as the beginning of a new cycle (renovation).”
Additional comments:
- The authors in their study aim to analyze the changes that have occurred in the physical (Body size and composition) characteristics of Italian female water polo athletes participating in the Olympic Games, from their first inclusion in the Olympics (2004) until today. Although they use physical characteristics the more accurate should be body size and composition, could be consider for same a matter of semantic.
ACCEPTED: modified as suggested: physical characteristics changed in body size and composition in all the manuscript.
- The topic is relevant considering the longitudinal data provided by the authors in a specific sport where the gender of the study is female, which is scarce on published data.
Thank you for your consideration.
- The used methodology is according to most of the studies published in this topic. However, it is missing some anthropometric variables normally used, and regarding BIA, it is unclear if the same equipment was used for all the Olympic teams and in the same conditions.
ANSWER: Unfortunately not all anthropometric variables were collected during the athletes’ evaluation. “All data were collected by the same operator using the same measurement protocol and equipment.” (line 130-131). The operator is one of the authors.
- The conclusions are consistent with the presented results and answer the main question.
Thank you for your consideration.
- The used references are adequate to the topic and adequate.
Thank you for your consideration.
- The presented tables are adequate; table 2 only establishes data from 2004 and 2024, and the other years? Was it the author's choice?
ACCEPTED: see point 2.
Reviewer 3 Report (New Reviewer)
Comments and Suggestions for Authors
The data presented in the manuscript can be of an interesting contribution to the literature, the authors are commended for their idea to examine secular trend changes in female water polo. However, the submitted manuscript still contains track changes of previous editing, I would expect more attention and careful work from the authors when finalizing the manuscript.
Specific comments:
Abstract:
Ln22: water polo players on the National Team...Please check for grammar (not correct use of “on”)
Ln23-24: a total number of 93 athletes were analyzed, for all 6 Olympics and for each examined Olympic. As stated, it can be misleading, please consider rephrasing.
Introduction
Reference to the hydration status seem not well introduced and explained. In addition, the connection of hydration status to performance was not examined in this study, therefore, we cannot really say anything about this. I recommend removing it.
Methods
It is stated that body height and body mass were measured twice and in case of a difference once more. Was this the case for all 6 examined Olympics? Aka, 20 years ago in Athens, athletes were measured two or even three times? Furthermore, what was the difference when a third measurement was required? Please explain for each relevant variable. Did you calculate test-retest reliability, and if not, why not?
Measurements were performed from the same operator for the entire period (20 years)?.
Similarly, BIA device was the same for all measurements ever 20 years? Please specify.
Results
Ln175: The expression “various editions of the Olympic Games” does not sound academic, please reconsider.
Ln176: I do not agree that standard deviation of mean age is rather low, coefficient of variation (CV) values range from 12% to even 21%. How did the authors decide sd being low?
Age secular decrease is not so convincing considering that in Tokio mean age was almost identical with that in Athens. Please reconsider.
Relative FFM (%) did not change, thus changes in absolute FFM are rather attributed to the increase in body mass. What does this mean, this has not been discussed at all in the manuscript.
It is not clear why relative age results were not reported for all six Olympics. It would be useful to see all data here, even if there were not any difference. But it seems that there was no statistical analysis used for relative age comparisons (although it would be necessary). Accordingly, how did the authors define significant difference between the 2004 and 2024 Olympics?
Discussion
Ln272-274: The connection between younger mean age with relative age is very speculative there is no analysis to support this. Please reconsider, a more clear and convincing explanation is needed here.
Ln309: BMI values confirm an optimal body mass in relation to body height. Its use as an indicator of good health is oversized. Please, consider rephrasing.
Ln328-335. Please be consistent with the use of acronyms for phase angle (PA or PhA). How did the authors explain that despite the increase in FFM (along with relatively constant body height) PhA did not change, assuming that FFM has the largest contribution to phase angle?
Ln350: how does hydration confirm good level of training, please provide an explanation.
Ln363: please add a reference supporting this statement.
Comments on the Quality of English LanguageEnglish language proof reading is recommended.
Author Response
Reply to reviewer #3
Comments and Suggestions for Authors
- The data presented in the manuscript can be of an interesting contribution to the literature, the authors are commended for their idea to examine secular trend changes in female water polo. However, the submitted manuscript still contains track changes of previous editing, I would expect more attention and careful work from the authors when finalizing the manuscript.
ANSWER: Thank you for your appreciation and feedback. The reason why there are track changes of previous editing is that at the time of re-submission it was not clear whether the article would be reviewed only by the Academic Editor and previous reviewers or by new reviewers. Therefore we gave the article to the journal with the revisions made, ready to edit where necessary.
Specific comments:
Abstract:
- Ln22: water polo players on the National Team...Please check for grammar (not correct use of “on”)
ACCEPTED: typing error. Corrected. “Data were collected from the female water polo players of the National Team…”.
- Ln23-24: a total number of 93 athletes were analyzed, for all 6 Olympics and for each examined Olympic. As stated, it can be misleading, please consider rephrasing.
ACCEPTED: modified. “A total of 93 athletes were assessed. Analyzed each series of the Olympics between 2004 and 2024.”
Introduction
- Reference to the hydration status seem not well introduced and explained. In addition, the connection of hydration status to performance was not examined in this study, therefore, we cannot really say anything about this. I recommend removing it.
ACCEPTED: removed.
Methods
- It is stated that body height and body mass were measured twice and in case of a difference once more. Was this the case for all 6 examined Olympics? Aka, 20 years ago in Athens, athletes were measured two or even three times? Furthermore, what was the difference when a third measurement was required? Please explain for each relevant variable. Did you calculate test-retest reliability, and if not, why not?
ANSWER: In all series of the Olympics, including Athens, the measurements assessed did not report different results and therefore the third test was never carried out.
- Measurements were performed by the same operator for the entire period (20 years)?.
ANSWER: Yes, the operator is one of the authors of this paper.
- Similarly, was the BIA device the same for all measurements over the 20 years? Please specify.
ANSWER: Yes, considering that these measurements were intended to provide data for a research project on secular trends, the operator took care to always use the same device.
Results
- Ln175: The expression “various editions of the Olympic Games” does not sound academic, please reconsider.
ACCEPTED: modified. “to participate in the series of the Olympic Games from 2004 until 2024.”
- Ln176: I do not agree that standard deviation of mean age is rather low, coefficient of variation (CV) values range from 12% to even 21%. How did the authors decide on the standard deviation being low?
ACCEPTED: this was an error in the text. Modified: “The groups’ average age has decreased over time, whereas, body mass, on the contrary, has increased.”
- Age secular decrease is not so convincing considering that in Tokyo the mean age was almost identical to that in Athens. Please reconsider.
ANSWER: even if data does not follow an absolute decrease (as highlighted in lines 177-178), the statistical trend line indicates a decreasing trend.
- Relative FFM (%) did not change, thus changes in absolute FFM are rather attributed to the increase in body mass. What does this mean, as it has not been discussed at all in the manuscript?
ACCEPTED: modified in results: “With regard to body composition, the FFM values exhibit an increasing trend also if relative FFM (%) does not change.”
Modified in discussion: “With regard to the FFM values (Table 1), we observed a significant increase (p = 0.05) due to the body mass increasing since relative FFM (%) does not change; this probably indicates that there was an increase in the need for muscle strength in competition and that muscle conditioning between the Athens (95% CI; 49.9-55.5) and Paris (95% CI 53.4-63.1) Olympic Games has maintained great importance. Undoubtedly, the characteristics of the game, which is extremely challenging in itself and increasingly demanding in terms of power, speed and rhythm, have led coaches to select athletes according to different criteria (more body mass) and…”
- It is not clear why relative age results were not reported for all six Olympics. It would be useful to see all data here, even if there were no differences. But it seems that there was no statistical analysis used for relative age comparisons (although it would be necessary). Accordingly, how did the authors define significant difference between the 2004 and 2024 Olympics?
ACCEPTED: As suggested we have reviewed the data and corrected some comments in the text. We have added a more detailed description of the methods used for the statistics, the relative age table has been modified, some sentences related to relative age in discussion have been modified.
Discussion
- Ln272-274: The connection between younger mean age with relative age is very speculative there is no analysis to support this. Please reconsider, a clearer and more convincing explanation is needed here.
ACCEPTED: thank you for the observation, which is correct and allows us to improve the manuscript: the connection between younger mean age with relative age is absolutely speculative and we modified the sentence: “These values show that younger athletes tend to be selected and that even if the data are not statistically significant, attention must be paid to a possible effect of relative age.”
- Ln309: BMI values confirm an optimal body mass in relation to body height. Its use as an indicator of good health is over-sized. Please, consider rephrasing.
ACCEPTED: modified: “The BMI values confirm an optimal body mass in relation to body height and show an increase consistent with that of body mass.”
- Ln328-335. Please be consistent with the use of acronyms for phase angle (PA or PhA). How did the authors explain that despite the increase in FFM (along with relatively constant body height) PhA did not change, assuming that FFM has the largest contribution to phase angle?
ACCEPTED: the acronyms have all been corrected to PhA. Phase angle in sports has been studied extensively in recent years, but we are still far from having perfectly defined scientific evidence on the factors determining its value or on the correlations with other performance variables.
Associations with ECW, ICW, ECW/ICW ratio, TBW, body cell mass, cellular integrity, age, mineral content, performance level and sport practised have been described (Campa F, Matias CN, Marini E, Heymsfield SB, Toselli S, Silva AM. Identifying Athlete Body-Fluid Changes During a Competitive Season With Bioelectrical Impedance Vector Analysis. Int J Sports Physiol Perform. 2019;11:1-7.
Lukaski HC, Kyle UG, Kondrup J. Assessment of adult malnutrition and prognosis with bio-electrical impedance analysis: phase angle and impedance ratio. Curr Opin Clin Nutr Metab Care. 2017;20:1–10). It is certainly a measure characterized by high variability whose applicability we still do not fully understand (Di Vincenzo O, Marra M., Scaffi L. Biolectrical impedance phase angle in sport: a systematic review. Journal of the International Society of Sport Nutrition 2019; 16: 49)).
Our study, due to its characteristics, is certainly not able to resolve these doubts and moreover it was not among the aims of our research. However, it could be hypothesized that, as in the athletes involved in the study, ECW and ICW do not change, so perhaps it is for this reason we did not find significant differences in PhA. Further scientific studies are necessary on this topic. We have added a plausible hypothesis to the text when explaining our data.
- Ln350: how does hydration confirm good level of training, please provide an explanation.
ACCEPTED: modified “The hydration values also confirm the good level of nutrition in the two groups..”
- Ln363: please add a reference supporting this statement.
ACCEPTED: a reference has been added [6].
Comments on the Quality of English Language
- English language proof reading is recommended.
ACCEPTED: the manuscript has been edited by an English language native.
Round 2
Reviewer 1 Report (New Reviewer)
Comments and Suggestions for Authors
In this version, the authors have implemented the corrections in accordance with the reviewer’s suggestions. I have only two comments. Minor revision – 2 comments. Please see the PDF file.

Author Response
Reply 2 to reviewer #1
- Delete reference: ACCEPTED: reference has been deleted.
Discussion:
- Point out the practical and theoretical purpose of the obtained results. Who can the obtained results serve? Added: “The results of this study may provide guidance to female water polo technicians in talent detection and selection.”
Reviewer 3 Report (New Reviewer)
Comments and Suggestions for Authors
Introduction
- In the manuscript please refer to body height (instead of height)
- Please consider rephrasing research hypothesis (Ln105-106) to sound more professional. I guess you assumed that similarly to other sports, body mass and body height increased over the examined period.
Methods
- Since measurements were conducted two times, It would be useful to see the results for test-retest reliability.
- The authors report a sample size of 93 athletes. Did you only include new players for each examined Olympics or were there players participated in more than one Olympics, please specify.
Please specify in statistical analysis:
- the factors used in the multifactorial ANOVA
- that Cohen d was used in pairwise comparisons between the 2004 and 2024 Olympics.
Results:
- Please use past tense to describe the results.
- Please include the results of normality tests. Were all variables normally distributed in all 6 Olympics?
- Please report ANOVA results for tables 1 and 3 and chi2 results for table 2. (Please include in the manuscript).
Please check In-text citations using format according to the journal’s instructions
Author Response
Reply 2 to reviewer #3
Introduction
- In the manuscript please refer to body height (instead of height). ACCEPTED: modified in all the manuscript as indicated.
- Please consider rephrasing research hypothesis (Ln105-106) to sound more professional. I guess you assumed that similarly to other sports, body mass and body height increased over the examined period. ACCEPTED: modified “The research hypothesis was that similarly to other sports, body mass and body height increased over the examined period.”
Methods
- Since measurements were conducted two times, It would be useful to see the results for test-retest reliability. ACCEPTED. Added: “In all athletes involved in the study, two consecutive measurements of body weight were performed and in case of a difference between the first and second measurement, a third measurement was performed. In case of a difference, the recorded weight was the average value of the three measurements. In our sample, the difference between the first and second measurement was very low (effect size: 0.02), not statistically significant (p: 0.63) and the repeatability between the tests was high (r: 0.99). A third measurement was used in 22% of the sample. The average difference between the first measurement (71.5 kg ± 10.0 kg) and the second measurement (71.7 kg ± 10.2) is very small.”
- The authors report a sample size of 93 athletes. Did you only include new players for each examined Olympics or were there players participated in more than one Olympics, please specify. ACCEPTED. Added “Athletes participating in more than one Olympic Game were included in the study only in their first participation.”
Please specify in statistical analysis:
- the factors used in the multifactorial ANOVA. ANSWER: “The factors used in ANOVA analysis were the different editions of the Olympics and the variables analyzed in the study”
- that Cohen d was used in pairwise comparisons between the 2004 and 2024 Olympics. ACCEPTED. The Statistics section has been modified as follows:
“Data were initially entered into an Excel database (Microsoft, Redmond, Washington, United States), and the subsequent analysis was performed using the Statistical Package for the Social Sciences Windows, version 19.0 (SPSS, Chicago, Illinois, USA), with descriptive statistics consisting of the mean ± standard deviation (SD). We evaluated the normality of data using normality plots, the Kolmogorov–Smirnov test and the Shapiro–Wilk test. The homogeneity of variance was assessed using Levene's test. The groups were compared using multifactorial univariate ANOVA considering the variables divided by the combined factor by year with the Bonferroni multiple comparisons option. Considering the sample size, standard deviation was used to calculate the pooled effect size. Cohen's d effect size (ES) was used to study the effect size according to the formula (M1-M2)/SDi, with M1 being the mean value of the first measurement, M2 being the mean value of the second, and SDi being the standard deviation for independent groups [16]. An effect size of 0.2 was considered small, 0.5 was considered medium, and 0.8 was considered large. The value of significance was set at 0.05. For non-normal parameters, comparisons were made with non-parametric tests. Comparisons between enumeration data were instead evaluated with the Chi2 test or, if there were frequencies less than 5, with the Fisher's exact test. Furthermore, the regression analysis was conducted by determining the best fitting quadratic curves for which the equations and determination coefficients (R2) are reported. A value of p<0.05 was considered statistically significant.”
Results:
- Please use past tense to describe the results. ACCEPTED: modified.
- Please include the results of normality tests. Were all variables normally distributed in all 6 Olympics? ANSWER: We believe that introducing this additional amount of data into the text weighs down the article and makes it less readable. Articles on secular trend do not report this type of data (Borms 2003 and Sedeaud et al 2014). The normality of all variables was tested with the statistical tools indicated in the Statistics section modified as described above: all data had a normal distribution. Only after we were sure of that we proceeded with the further statistical analysis. We followed a rigorous methodological and statistical approach: all data are accessible and available at the Statistical Center of the University of Rome Tor Vergata where the statistical analysis was performed.
- Please report ANOVA results for tables 1 and 3 and Chi2 results for table 2. (Please include in the manuscript). ANSWER: See above.
Please check In-text citations using format according to the journal’s instructions. ACCEPTED: Having checked the format of in-text citations we were unable to find differences with the journal’s instructions. Could you kindly indicate the errors?
Round 3
Reviewer 3 Report (New Reviewer)
Comments and Suggestions for Authors
Please check in Ln 28 and consider correcting to: Their body do not…
Author Response
Comment 1: Please check in Ln 28 and consider correcting to: Their body do not…
ACCEPTED. Modified: "Their body height do not vary..."
This manuscript is a resubmission of an earlier submission. The following is a list of the peer review reports and author responses from that submission.
Round 1
Reviewer 1 Report
Comments and Suggestions for Authors
I found this article on high-level female water polo players addressing the scarcity of studies on this topic very interesting.
To enhance the paper, I would like to discuss some basic concerns and recommendations.
The paper needs to be edited by an English editor or a native English individual. Through this editing, the article’s value will be elevated, and specific points will be clarified.
Although the paper is innovative and tracks long-term changes, it is important to have the term “Italian” in the title and inside the article. The growth of Italian female water polo athletes is being observed in this study for a certain duration.
Nevertheless, this does not accurately reflect the overall advancement of female water polo players. When it comes to water polo, countries with top rankings display dissimilar body compositions in comparison to Mediterranean countries like Italy, Greece, and Spain.
This localization does not undervalue the importance of the present work. In contrast, it encourages further research on the subject.
Here are a few suggestions you might want to take into account.
Abstract
Line 25. Include (BC) abbreviation in sentence in line 25. All abbreviations in the paper follow the same pattern. Please refer to the full description and abbreviation (e.g. FFM) when mentioned for the first time.
Line 27. BMI confirms that the athletes were always healthy and properly fed.
What else BMI may display about the body image, for example?
24.9 marks the upper threshold of healthy and the beginning of obesity. Additionally, according to SD, some athletes had a body mass index above 24.9.
Line 31. The most likely explanation.
The existence of various international competitions for teenage female water polo players (European, world, etc.) could be seen as further evidence of this evolution.
Introduction
When referring to hydration results, please provide information on its importance, measurement methods, and other relevant details.
Similarly, for phase angle (PhA).
Methods
- Can you provide information on BIA, highlighting its distinctions from other methods and noting any limitations?
- Please could you refer the Cohen's d effect size ranges?
- From my perspective, you should emphasize important findings on your tables.
Discussion
Discussion must be edited for clarity and specificity.
Conclusion
Please rephrase the sentence “Better anthropometric’. Maybe more suitable to new game style?
Comments on the Quality of English LanguageThe paper needs editing to ensure clarity and formal English language. The article’s value will be increased, and specific points will be clarified through this editing.
Reviewer 2 Report
Comments and Suggestions for Authors
The paper analyzes body composition and age over 20 years for female water polo players participating in the Olympic games from 2004 to 2024.
The paper is well written overall and the analysis looks sound. My biggest concern is that the intention of the paper itself and following conclusion isn’t of particular interest to the readers. Slightly more muscle mass at an early age is a general trend for most sports as the science on strength and condition improves over the years. I don’t see this paper bringing significant value to the reader with its general conclusions.
Is there an issue around BMI with water polo players? Is there malnourishment or overtraining? Is there anything that can be concluded or extracted based on this data that is useful for other researchers, coaches etc.? Also, comparing data only between 2004 and 2024 is convenient, as these two years seem to be the outliers. The years in between are much closer in data points and would probably not produce any significant results.
Other minor comments:
Abstract:
FFM, line 28, should be written out as the abbreviation wasn’t used in the text yet.
Methods:
Which University IRB approved the study? Please specify as there are 6 different institutions listed with the authors of this paper. The IRB number of the study should be listed as well.
BMI and BIA were already explained in previous sections, no need to repeat.
BMI formula is incorrect (line 99), height must be squared.
Figure 2: all abbreviations need to be explained in the caption (FM, FFM, PhA), same goes for Table 2. Figures and Tables need to be stand alone, just like you did with Table 1.
Comparing 2004 and 2024 shows some significant differences although the trend isn’t so clear when looking at other years. It would be interesting to see comparison between 2012 or 2016 and 2024. I’m not sure if similar significant age decrease would be observed from this data. Therefore, it is presumptuous to make such bold statements in the discussion section. I’m not sure such conclusions can be drawn based on this relatively small and single nationality sample size.
Reviewer 3 Report
Comments and Suggestions for Authors
The Authors analyzed changes in body composition and anthropometric data among water polo female athletes. Specifically, the recruited sample (N=93) consisted of athletes participating in the Olympic Games from 2004 to the present. Results showed a reduction in the athletes’ age over time and an increase in body mass; height did not change instead. The Authors concluded that the athletes recently recruited for the Olympic Games had more favorable body composition and anthropometric characteristics as compared to the athletes from previous Olympic Games.
Although the manuscript has overall merit, the scientific approach to writing requires extensive revisions. Therefore, the English quality of the manuscript should be improved.
I also provide some minor comments as follows.
Article title: letters should be capitalized (Olympic Female Water Polo Players…)
Abstract: cite FFM in full before using the abbreviation; throughout the abstract semicolon “;” should be removed after headings.
Introduction: this section could be improved by adding some references and a more in-depth overview of water polo, providing more scientific background and linking the aim of the study. At the moment, this section provides minimal and only superficial information.
Line 46: consider changing “fat free mass”
Line 47 and 57: cite in full BIA and BC and then use the abbreviation
Tables: it would be helpful adding in the footnotes that results are showed as mean ± standard deviation (SD)
Table 2: why the authors use the arbitrary unit to describe the BMI? Kg/(m)2 could be used instead. Also, “a.u.” should be added in the footnotes.
Table 3: data in columns need to be properly aligned
Table 5: consider changing the “,” comma with “.” the full stop to indicate the decimal numbers
Discussion: line 240-241, please revise the in-text citation
References: numbers should be removed in ref. no. 18 to 20. References should be re-formatted according to Journal’s guidelines.
Comments on the Quality of English LanguageThe scientific approach to writing requires extensive revisions. Therefore, the English quality of the manuscript should be improved.
Round 2
Reviewer 2 Report
Comments and Suggestions for Authors
The journal provides clear instructions on how to respond to reviewers with templates. "A part has been inserted." is an insufficient response. You need to provide specific text that was inserted and the location (line numbers) for each response.
I don't think the paper has improved greatly since the last revision.
Author Response
- The paper is well written overall and the analysis looks sound. My biggest concern is that the intention of the paper itself and following conclusion isn’t of particular interest to the readers. Slightly more muscle mass at an early age is a general trend for most sports as the science on strength and condition improves over the years. I don’t see this paper bringing significant value to the reader with its general conclusions. ANSWER: We completely agree with the general trend in muscle mass but we wanted to demonstrate whether this trend is also real in female water polo players. The value provided to the reader is in the data regarding female water polo players that did not exist before.
- Is there an issue around BMI with water polo players? Is there malnourishment or overtraining? Is there anything that can be concluded or extracted based on this data that is useful for other researchers, coaches etc.?
ACCEPTED: has been inserted “We only considered BMI in this context because, in athletes, it must be used with caution; this is due to the greater muscle mass of athletes and the use of criteria that are different to those used in the normal population [19].” Line 293-296.
Actually, we used BMI mainly to identify the presence of malnourishment, sometimes induced in women by body image problems.
- Also, comparing data only between 2004 and 2024 is convenient, as these two years seem to be the outliers. The years in between are much closer in data points and would probably not produce any significant results.
ANSWER: In our article we wanted to study the secular trend and not the wave between time. To study a trend, as other authors in other sports have done, we chose a time span of 20 years against shorter periods of only 4 years. Furthermore, after 4 years the two teams analyzed could have had some athletes in common, thus making the result falsely more homogeneous and hiding the real differences brought about by the elapsed time.
Other minor comments:
Abstract:
FFM, line 28, should be written out as the abbreviation wasn’t used in the text yet. ACCEPTED: corrected as suggested. Line 31.
Methods:
- Which University IRB approved the study? Please specify as there are 6 different institutions listed with the authors of this paper. The IRB number of the study should be listed as well.
ACCEPTED: has been inserted “This study was approved by the Internal Review Board of the University of Tor Vergata, with number 135 and dated 10/06/2024, and it followed the guidelines of the 1964 Declaration of Helsinki and subsequent amendments”. Line 123-125.
- BMI and BIA were already explained in previous sections, no need to repeat.
ACCEPTED: corrected as suggested Line 130 and 136.
- BMI formula is incorrect (line 99), height must be squared.
ACCEPTED: corrected as suggested “BMI was calculated as body mass (kg) / height (m)2”. Line 134.
Figure 2: all abbreviations need to be explained in the caption (FM, FFM, PhA), same goes for Table 2. Figures and Tables need to be stand alone, just like you did with Table 1. ACCEPTED: corrected as suggested.
5a. Comparing 2004 and 2024 shows some significant differences although the trend isn’t so clear when looking at other years. It would be interesting to see comparison between 2012 or 2016 and 2024. I’m not sure if similar significant age decrease would be observed from this data.
ANSWER: Analyzing a period shorter than 20 years (4 or 8 years) there was a risk that some athletes could be the same and therefore make the samples more similar. After 20 years this risk no longer exists. Authors who studied the secular trend have also considered much longer periods.
5b. Therefore, it is presumptuous to make such bold statements in the discussion section. I’m not sure such conclusions can be drawn based on this relatively small and single nationality sample size.
ACCEPTED: It was added in the limitations of the article that the study essentially concerned Italian athletes and so the results are to be considered limited to a Latin population.
We are grateful to the reviewer for his contribution which, together with that of the other reviewers, helped us to improve the article, adding various parts and correcting some format errors, as can be seen by viewing the corrections made in the new text.

Reviewer 3 Report
Comments and Suggestions for Authors
The manuscript has been revised as requested. I have no more comments.
Author Response
Thank you very much for your contribution.